# Long-Term Survival of Neuroblastoma Patients Receiving Surgery, Chemotherapy, and Radiotherapy: A Propensity Score Matching Study

**DOI:** 10.3390/jcm12030754

**Published:** 2023-01-17

**Authors:** Qilan Li, Jianqun Wang, Yang Cheng, Anpei Hu, Dan Li, Xiaojing Wang, Yanhua Guo, Yi Zhou, Guo Chen, Banghe Bao, Haiyang Gao, Jiyu Song, Xinyi Du, Liduan Zheng, Qiangsong Tong

**Affiliations:** 1Department of Pediatric Surgery, Union Hospital, Tongji Medical College, Huazhong University of Science and Technology, 1277 Jiefang Avenue, Wuhan 430022, China; 2Clinical Center of Human Genomic Research, Union Hospital, Tongji Medical College, Huazhong University of Science and Technology, 1277 Jiefang Avenue, Wuhan 430022, China; 3Department of Pathology, Union Hospital, Tongji Medical College, Huazhong University of Science and Technology, 1277 Jiefang Avenue, Wuhan 430022, China

**Keywords:** neuroblastoma, surveillance, epidemiology and end results, surgery, chemotherapy, radiotherapy, long-term survival

## Abstract

Neuroblastoma is the most common extracranial solid malignancy in children. This study was undertaken to determine the long-term survival of neuroblastoma patients receiving conventional therapeutics (surgery, chemotherapy, and radiotherapy). The neuroblastoma patients examined were registered in the Surveillance, Epidemiology and End Results (SEER) database (1975–2016). Using propensity score matching analysis, the patients were paired by record depending on whether they received surgery, chemotherapy, or radiotherapy. Univariate and multivariate analyses of the disease-specific survival of the paired patients were performed by the log-rank test and Cox regression assay. A total of 4568 neuroblastoma patients were included in this study. During 1975–2016, the proportion of histopathological grade III/IV cases receiving surgery gradually increased, while the number of patients with tumors of grade I to IV undergoing chemotherapy or radiotherapy was stable or even decreased. After propensity score analysis, for Grade I + II and Grade III tumors, surgery obviously improved the disease-specific survival of patients, while chemotherapy was unfavorable for patient prognosis, and radiotherapy exerted no obvious effect on the patients. However, no matter what treatment was chosen, the patients with advanced-histopathological-grade tumors had a poor prognosis. Meanwhile, for all histopathological grades, the patients receiving surgery and subsequent chemotherapy or radiotherapy suffered from worsen disease-specific survival than those simply undergoing surgery. Fortunately, the negative effects of surgery, chemotherapy, or radiotherapy improved gradually over time. Surgery improved the long-term survival of the neuroblastoma patients, while chemotherapy and radiotherapy exerted an unfavorable impact on patient outcome. These results provide an important reference for the clinical treatment of neuroblastoma.

## 1. Introduction

Neuroblastoma is the most common extracranial solid malignancy in childhood and comprises 8–10% of all pediatric cancers [1]. Neuroblastoma primarily occurs in the adrenal gland, retroperitoneum, neck, chest, and other organs [1]. Based on prognostic factors such as age older than 18 months, histopathology, and MYCN amplification, neuroblastoma patients are stratified into different risk groups [2] and treated according to different strategies: (I) the low-risk group is mainly treated with surgery; (II) the intermediate-risk group is managed with surgery combined with moderate-intensity chemotherapy; (III) the high-risk group is treated with surgery, chemotherapy, radiotherapy, autologous hematopoietic stem cells, and immunotherapy; (IV) patients with stage 4S tumors are mainly managed by supportive therapy [3,4]. However, the association of demographic or clinicopathological factors of neuroblastoma patients with therapeutic efficacy remains to be further determined. 

In 2015, the 10-year overall survival rate for stage 1–3 neuroblastoma patients was approximately 91%, while that for stage 4 cases older than 18 months was 38% [5]. As an effective treatment for neuroblastoma, surgery significantly improves patients’ quality of life and prolongs their survival [6]. In high-risk cases, preoperative chemotherapy reduces the tumor volume and decreases the intraoperative bleeding or surgical risk [7,8]. In combination with biotherapy, chemotherapy also increases the 3-year survival of neuroblastoma patients [9]. However, although higher remission rates are noted with an intensive induction chemotherapy regimen combined with surgical resection or radiotherapy, patient prognosis and long-term survival are only modestly improved in less than one-third of patients [10,11] and remain dismal in cases with metastatic neuroblastoma [9,12]. Therefore, the impact of these conventional therapeutics on neuroblastoma patients’ long-term survival warrants further investigation. 

In this study, we performed an analysis of neuroblastoma cases registered in the Surveillance, Epidemiology and End Results (SEER) public-access database for the period from 1975 to 2016, which were collected from various geographic areas in the United States [13]. By using propensity score matching analysis to rule out the potential impact of demographic (age, gender, and race/ethnicity) or clinicopathological (year of diagnosis, primary site, and histopathological grade) factors, we discovered that surgery significantly increased the patients’ long-term survival. Unexpectedly, chemotherapy or radiotherapy reduced the disease-specific long-term survival of the neuroblastoma patients, including those treated with surgery. The combination of surgery, chemotherapy, and radiotherapy is not more effective than surgery alone in improving patient survival, suggesting the importance of exploring more effective treatments for neuroblastoma.

## 2. Materials and Methods

### 2.1. Data Source

The SEER program of National Cancer Institute (NCI) collects information on demographic (age, gender, race/ethnicity), diagnostic, anatomical, histological, therapeutic, and survival features of neuroblastoma patients (https://seer.cancer.gov, accessed on 21 April 2020). The long-standing SEER 18 registries (Alaska, Atlanta, Connecticut, Detroit, Greater California, Greater Georgia, Hawaii, Iowa, Kentucky, Los Angeles, Louisiana, New Jersey, New Mexico, Rural Georgia, San Francisco-Oakland, San Jose-Monterey, Seattle-Puget Sound, and Utah) contain epidemiologic data covering approximately 34.6% of the United States population. Individuals with neuroblastoma (International Classification of Childhood Cancer (ICCC) codes 9490 and 9500, third edition) diagnosed in the period 1975–2016 were selected through SEER*Stat software (version 8.1.5). The individuals were classified by age at diagnosis (less than 1, 1 to 3, 4 to 6, 7 to 14, 15 to 18, and older than 18 years), while the year of diagnosis was applied as a variable, with five periods (1975 to 1984, 1985 to 1994, 1995 to 2004, and 2005 to 2016). Since the race records reported mainly whites and blacks, the remaining ethnicities were classified as one group. In addition, since most neuroblastomas occurred in the adrenal gland and retroperitoneum (56.4%), other primary sites were classified as one group. 

### 2.2. Patient Cohort

We retrieved 5003 neuroblastoma patients in SEER*stat by excluding those with olfactory neuroblastoma. Patients with multiple tumors or who died from other cause were further excluded, resulting in retained 4568 cases with a histochemical or pathological diagnosis as well as full records of vital status and survival time. Then, using treatment methods as indicators for propensity score matching, 974, 1064, and 980 pairs of cases were successfully matched for comparing the long-term survival of patients with neuroblastoma receiving surgery, chemotherapy, or radiotherapy, respectively.

### 2.3. Survival Outcome and Study Variables

This study mainly evaluated the effects of three types of treatment (surgery, chemotherapy, and radiotherapy) on the long-term disease-specific survival (DSS) of neuroblastoma patients, as well as the influence of demographic (age, gender, and race/ethnicity) or clinicopathological (year of diagnosis, primary site, and histopathological grade) factors on patients’ outcome. Survival was measured from the date of diagnosis to the date of death from the disease or the last contact. The endpoint was late neuroblastoma mortality and was determined by the disease-specific vital status and survival time of the patients. 

### 2.4. Propensity Score Matching Analysis

Propensity score matching (PSM) analysis was used to mitigate discrepancies related to nonrandom selection by R (version 4.1.0; Vienna, Austria); treatment modalities (surgery, chemotherapy, or radiotherapy) were applied as indicators, and baseline variables (age, gender, race, year of diagnosis, primary site, histopathological grade, and other treatment modalities except for the indicators) were considered as covariates, for 1:1 nearest-neighbor propensity score matching analysis using a logistic regression model and a caliper width of 0.01, a method capable of replicating randomized controlled trial results for observational studies. The model was demonstrated to be adequate based on the pairwise comparison of the matched variables (Table 1).

### 2.5. Statistical Analysis

The baseline characteristics were compared by χ^2^ analysis for the categorical variables and the Student’s *t*-test for the continuous variables. Univariable and multivariable models were created by using Cox proportional hazards for time-related data. The ‘Forward Wald’ stepwise procedure was applied to build the multivariable Cox regression model. Variables with *p*-values < 0.05 in univariate analyses were candidates for multivariable Cox regression analysis. Kaplan–Meier analysis was used to visually display the survival time curves. Statistical analyses included the total number of patients in groups and were performed using SPSS version 18 (SPSS statistical software, Chicago, IL, USA). All statistical tests were two-way, and *p* values less than 0.05 were considered statistically significant. Trends and survival curves were drawn by GraphPad version 8.0 (GraphPad Software, San Diego, CA, USA).

## 3. Results

### 3.1. Characteristics of the Neuroblastoma Patients

This study included 4568 neuroblastoma cases registered from 1975 to 2016 in the SEER database; the detailed inclusion procedures are presented in Figure 1. There was a higher percentage of patients younger than 18 years (95.7%), of male gender (52.4%), white ethnicity (78.4%), and with a diagnosis after the year 1996 (70.6%, Table 2). The major (54.4%) of primary sites were adrenal gland and retroperitoneum (Table 2). Most patients received surgery (73.6%) or chemotherapy (64.9%), while a small portion of patients underwent radiotherapy (25.6%, Table 2). Notably, after multivariable Cox regression analysis, the DSS time was significantly longer for younger patients, with a primary tumor site different from the adrenal gland, a better histopathological grade, a more recent year of diagnosis, receiving surgery but not treated with chemotherapy or radiotherapy (Table 3).

### 3.2. Trends and Effects of Surgery, Chemotherapy, and Radiotherapy

Based on the trend statistics of these neuroblastoma patients from 1975 to 2016, we found that the proportion of histopathological grade I/II cases undergoing surgery did not significantly change, while that of grade III/IV cases undergoing surgery gradually increased and reached a maximum of 75% from 2000 to 2005 (Figure 2A). In contrast, the proportion of grade I to IV patients receiving chemotherapy was relatively stable, and that of patients receiving radiotherapy remained at a relatively low level, especially from 1995 to 2000 (Figure 2A). The DSS time for each histopathological grade was respectively compared on the basis of the conventional treatment received (surgery, chemotherapy, or radiotherapy) by using the data after PSM analysis, which eliminated the factors that might affect the DSS time, ensuring the outcome was only dependent upon whether the patients received a specific treatment. The log-rank test and Kaplan–Meier curves revealed a significantly longer DSS time for patients with Grade I + II [hazard (HR) = 10.04, *p* = 0.007] and Grade III [HR = 2.26, *p* < 0.001] tumors who underwent surgery, while chemotherapy was unfavorable for the prognosis of the patients in the Grade III group [HR = 0.21, *p* < 0.001]. Meanwhile, for the patients in the Grade I group, chemotherapy was associated with a trend of poor prognosis without a significantly statistical difference [HR = 0.30, *p* = 0.05]. Radiotherapy exerted no obvious effect on the patients in these two groups. As for the patients with an advanced histopathological grade (Grade IV) NB, no matter what treatment chosen, a poor long-time prognosis was noted.

### 3.3. Impact of Baseline Characteristics and Effects of Combinatorial Therapies on the Long-Term Survival of Neuroblastoma Patients Receiving Surgery

Since the above results revealed that surgery resulted in a favorable outcome for patients with neuroblastoma, we further explored the impact of demographic or clinicopathological factors on neuroblastoma patients receiving a surgical treatment with unmatched data. The results showed that the DSS time was significantly longer for younger patients receiving surgery, with a primary tumor site other than the adrenal gland, a better histopathological grade, a more recent year of diagnosis, but not treated with chemotherapy or radiotherapy (Table 4). Then, we compared the DSS time of the patients who simply received surgery or simultaneously received combinatorial therapies at different stages with unmatched data. The KM curve indicated that, regardless of the tumor stage, a purely surgical treatment without chemotherapy or radiotherapy led to a more satisfactory result than the combinatorial therapies (Figure 3).

### 3.4. Trend of the Conventional Treatment Results with the Advancement of Time

On account of the worsen outcomes of patients who received chemotherapy or radiotherapy as described above, we further investigated the trend of the conventional treatment results with the advancement of time. Of note, there was an increasingly favorable outcome for all three therapeutic methods with the advancement of time (Figure 4), which indicated an excellent effect of chemotherapy and radiotherapy associated with surgery for patients with neuroblastomas at advanced histopathological grades.

## 4. Discussion

Currently, most clinical studies focus on the short- or medium-term survival of neuroblastoma patients, while the roles of prognostic factors and treatments in the long-term prognosis of neuroblastoma patients remain unclear. In this study, we performed an analysis of demographic, clinical, and survival data from 18 SEER registries. In line with previous studies [14,15], we found that patient age, tumor primary site, and histopathological grade IV are risk factors for a worse outcome in neuroblastoma patients. Despite the short follow-up duration, a later diagnosis date was a favorable factor for the patients, suggesting a recent improvement of medical care. More importantly, surgery improved the long-term DSS of the neuroblastoma patients, while chemotherapy or radiotherapy appeared as unfavorable factors for patient survival. 

For low- and intermediate-risk neuroblastoma cases, surgery is recommended mainly to completely remove the tumor tissues, which reduces the possibility of metastasis and recurrence [16]. Considering tumor infiltration into blood vessels or important organs, surgery for high-risk neuroblastoma is performed to remove the tumor as much as possible so to maintain the patient’s quality of life [6]. A series of studies indicate that surgery is able to effectively improve the survival rate of low-, intermediate-, and high-risk neuroblastoma patients [17,18]. For stage 1 and stage 2 cases, surgery alone is an effective and safe treatment, and the administration of chemotherapy may be restricted to specific situations [19]. A retrospective institutional series of stage 3 patients also supports the ‘‘surgery alone concept” [20]. On the other hand, because of the efficacy of radiotherapy/chemotherapy, it seems likely that aggressive surgery is unnecessary in high-intensity multimodal treatments [21]. Previous studies showed no difference in the effects of total or subtotal resection on the short-term survival of high-risk patients [22], especially those aged 18 months or older suffering from metastatic neuroblastoma [23], also considering that serious complication or morbidity of technically difficult and long-lasting operations should be minimized [5]. Therefore, the impact of surgery on the long-term outcome of neuroblastoma patients is a matter of medical debate. In this study, based on propensity score matching analysis, we found that surgery consistently improved the long-term survival of neuroblastoma patients of early histopathological grade, regardless of chemotherapy or radiotherapy administration, suggesting its beneficial role in the clinical treatment of neuroblastoma at an early stage. In contrast, for patients with advanced histopathological grade, surgery did not achieve the ideal therapeutic efficacy and neither prolonged life expectancy in comparison to patients who did not receive surgery. Thus, more efficient treatment methods or combinations of methods deserve to be explored for these patients.

Chemotherapy is an important treatment method for intermediate- and high-risk neuroblastoma which kills the tumor cells, reducing the tumor size and decreasing the risks of surgery [24,25]. Infants with stage 4 disease are believed to require chemotherapy to be cured [26]. By using moderately aggressive chemotherapy, the overall survival rate of 80% can be achieved for intermediate-risk cases [27], while intensive chemotherapy also improves the 5-year overall survival rate of patients suffering from unresectable neuroblastoma [8]. Preoperative adjuvant chemotherapy is associated with a high disease-free survival rate (97.1%) but does not affect the 10-year survival of neuroblastoma patients. Compared with standard regimens, a rapid induction regimen with an increased dose intensity seemed to improve the 3-year (31.0% vs. 24.2%), 5-year (30.2% vs. 18.2%), and 10-year (27.1% vs. 18.2%) event-free survival of high-risk neuroblastoma patients. However, more infective complications and longer hospital stays have been documented in patients assigned a rapid regimen compared to those receiving a standard treatment. High-dose chemotherapy might cause late complications, including growth failure, renal dysfunctions, hypothyroidism, hearing impairment, orthopedic impairment, and secondary malignancies [24]. To our knowledge, no randomized trial has studied the effect of chemotherapy on the long-term survival of neuroblastoma patients. Based on propensity score-matching analysis of patients in the SEER database, we found that chemotherapy was an unfavorable factor for the long-term survival of neuroblastoma patients, including those who received surgery. This might be associated with the toxicity and impact of chemotherapeutic drugs on the development and immunity of children, which warrants further investigation. 

In the 1980s and 1990s, radiotherapy (20–40 Gy) was uniformly applied to neuroblastoma patients, resulting in late irradiation-related side effects such as scoliosis and growth abnormalities [28]. In recent years, low to moderate doses of radiotherapy are considered an important adjunct treatment for high-risk neuroblastoma patients, especially for the consolidating locoregional control of residual and relapsed tumors or for treating resistant metastatic tumors [29,30]. A study in South Africa showed that patients receiving radiotherapy had a high 2-year overall survival rate (44.8%) [31]. However, cancer survivors receiving low doses of radiotherapy in childhood have a 1.6-fold risk of developing cardiac disease over the next 30 years [32]. Although radiotherapy is cytotoxic to tumor cells, it also has deleterious effects in normal tissues, resulting in growth and developmental failure, gastrointestinal dysfunction, and pulmonary or cardiac abnormalities, which may lead to a poor long-term prognosis of neuroblastoma patients [33]. Hopefully, improvements in technologies allowing the delivery of lower cumulative doses in smaller volumes will result in fewer side effects. In this study, we found that radiotherapy shortened the long-term survival of neuroblastoma patients, suggesting caution when applying it in clinical therapies.

Patients with high-risk neuroblastoma have a poor prognosis, and survivors are often left with debilitating long-term sequelae from the treatments received. Prior to 2009, the therapy for high-risk neuroblastoma relied on surgery, local radiotherapy, and gradually more aggressive treatment combinations including chemotherapy regimens. While this approach prolonged the survival in some cases, fewer than 40% of the patients survived for more than 5 years without relapse; a relapsed disease could only rarely be cured [34]. After 2010, the anti-disialoganglioside-2 (anti-GD2) monoclonal antibody therapy emerged as a standard protocol. Although the 2-year overall survival of patients receiving the adjuvant anti-GD2 monoclonal antibody with interleukin-2 (IL-2), granulocyte-macrophage colony-stimulating factor (GM-CSF), and retinoic acid rose to 86%, approximately 50% of the patients would relapse and eventually die from their disease [35], and the actual 5-year overall survival rates were only about 50% [36], indicating a slightly improvement of the prognosis compared to that achieved with conventional treatments. A more available method for high-risk neuroblastoma is the adoptive transfer of chimeric antigen receptor (CAR) T cells, which has a good potential to be successful. In early-phase clinical trials, CAR-T cells therapy for neuroblastoma has proven safe and feasible, but significant barriers to its efficacy remain [37]. Although this study did not evaluate the effect of immunotherapy due to a lack of relevant records in the SEER data, we discussed other possible treatment methods which are beneficial for patients with advanced risk for whom the conventional treatments have limited effects.

Although chemotherapy and radiotherapy were associated with a worsen prognosis in patients with neuroblastoma, their effect improved gradually over time. Chemotherapy and radiotherapy may assist surgery by reducing the tumor size before a surgical treatment or suppressing tumor relapse after surgery. Thus, it is of great importance to attempt and formulate new regimens of chemotherapy and radiotherapy to improve the DSS of patients with neuroblastoma in an advanced stage, whose tumors cannot be simply removed by surgery.

There are some limitations of our analysis. Firstly, due to a lack of individual-level records, we could not obtain specific treatment details of the patients, including the surgical techniques and chemotherapy or radiotherapy plans adopted, which restricted our further evaluation of their impact on the long-term survival of the patients. Secondly, recent studies have shown that specific molecular characteristics of neuroblastoma are associated with tumor sub-categories [38,39] and clinical outcomes [34,40]. Although we applied stratification factors to reduce data bias, cytogenetic, molecular, or unknown factors related to patients’ survival were still unavoidably omitted. Thirdly, the tumor grade of some patients in the SEER data was unclear, which may have limited the statistical analysis results and reported conclusions. Additional studies on treatment variables are needed to further clarify their impact on the long-term survival of neuroblastoma patients. 

In conclusion, as an effective treatment for neuroblastoma, surgery appeared beneficial for the long-term survival of patients with a low-histopathological-grade tumor, regardless of chemotherapy or radiotherapy administration, while for patients with tumors of advanced histopathological grade, the conventional treatments did not influence effectively the long-time prognosis. Unexpected, chemotherapy or radiotherapy did not effectively improve the long-term survival of neuroblastoma patients, including those receiving surgery. We believe that this study extends our knowledge about the benefits and shortcoming of conventional therapeutics and provides an important reference for the choice of clinical treatments for neuroblastoma.

## Figures and Tables

**Figure 1 jcm-12-00754-f001:**
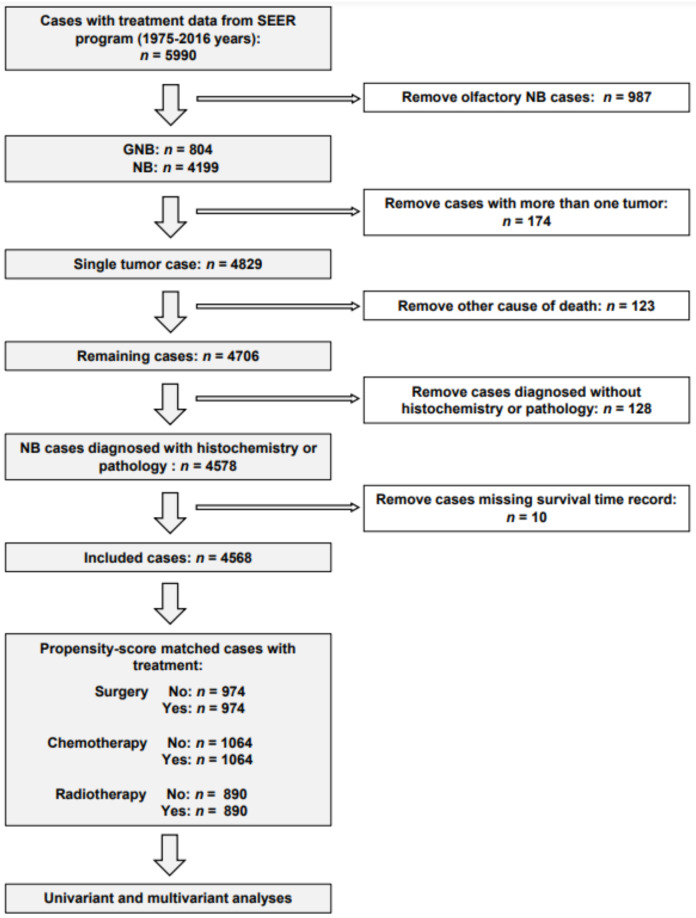
Process of screening of the studied data from the SEER database.

**Figure 2 jcm-12-00754-f002:**
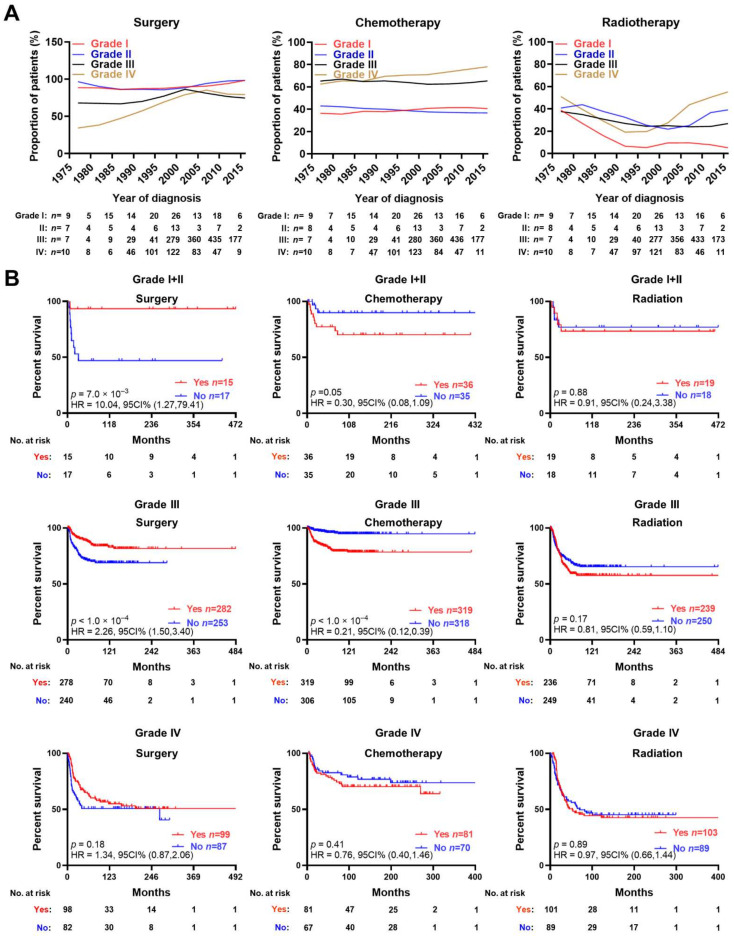
Trends and effects of surgery, chemotherapy, and radiotherapy. (**A**) Proportion of neuroblastoma patients with different histopathological grades undergoing surgery, chemotherapy, or radiotherapy during 1975–2016. (**B**) Kaplan–Meier curves reflecting the disease-specific survival of propensity score-matched neuroblastoma patients receiving surgery, chemotherapy, or radiotherapy, with tumor of different histopathological grades. Log-rank test for the survival comparison in (**B**).

**Figure 3 jcm-12-00754-f003:**
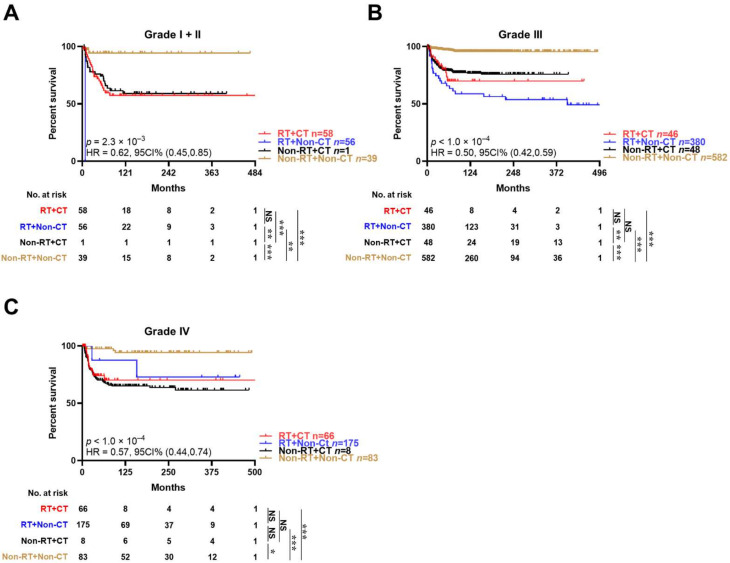
Effects of combinatorial therapies on the long-term survival of neuroblastoma patients receiving surgery, with tumors of different histopathological grades in unmatched data. (**A**) Patients with both grade I and grade II tumors; (**B**) Patients with grade III tumors; (**C**) Patients with grade IV tumors. Kaplan–Meier curves reflecting the disease-specific survival of patients who underwent the indicated treatment. CT, chemotherapy; RT, radiotherapy. Log-rank test for survival comparison in (**A**–**C**). * *p* < 0.05, ** *p* < 0.01, *** *p* < 0.001. NS, non–significant.

**Figure 4 jcm-12-00754-f004:**
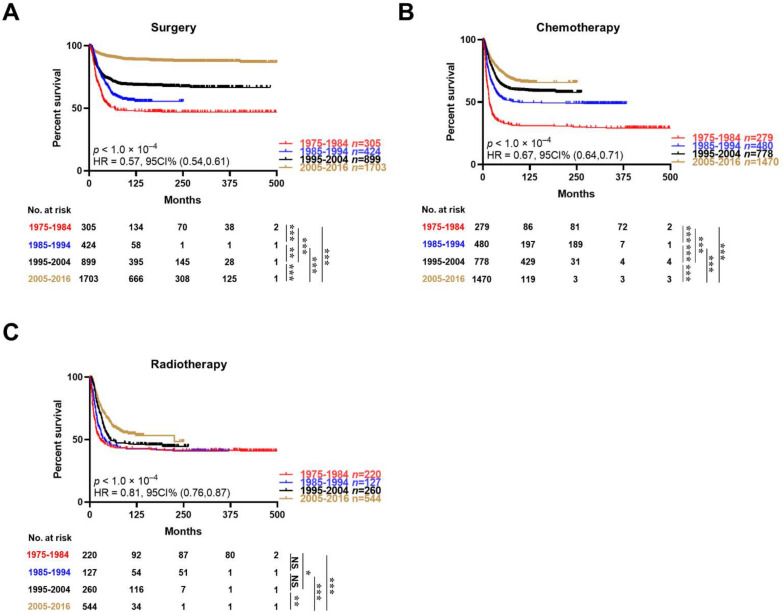
Trend of the conventional treatment results with the advancement of time in unmatched data. (**A**): Patients receiving surgery; (**B**) Patients receiving chemotherapy; (**C**): Patients receiving radiotherapy. Kaplan–Meier curves reflecting the disease-specific survival of patients who underwent the indicated treatment. Log-rank test for survival comparison in (**A**–**C**). * *p* < 0.05, ** *p* < 0.01, *** *p* < 0.001. NS, non-significant.

**Table 1 jcm-12-00754-t001:** Baseline characteristics before and after matching according to whether the patients received surgery, chemotherapy or radiation therapy.

	Unmatched			Matched			Unmatched			Matched			Unmatched			Matched		
	SG (*n* = 1147)	Non-SG (*n* = 3363)	*P*	SG (*n* = 1147)	Non-SG (*n* = 3363)	*p*	CT (*n* = 2966)	Non-CT (*n* = 1602)	*p*	CT (*n* = 1064)	Non-CT (*n* = 1064)	*p*	RT (*n* = 1169)	Non-RT(*n* = 3354)	*p*	RT (*n* = 890)	Non-RT (*n* = 890)	*p*
Gender			0.328			0.158			0.039			0.516			0.087			0.924
Female (%)	48.1	46.4		43.8	47.0		46.5	49.7		49.7	51.1		45.3	48.2		43.6	43.4	
male (%)	51.9	53.6		56.2	53.0		53.5	50.3		50.3	48.9		54.7	51.8		56.4	56.6	
Grade			<0.001			0.063			<0.001			0.485			0.545			0.682
I: well differentiated (%)	3.3	1.2		0.8	1.3		54.2	62.5		59.0	60.2		1.4	3.3		1.3	1.3	
II: moderately differentiated (%)	1.4	0.3		0.7	0.4		1.8	4.5		2.5	2.3		1.2	1.1		0.8	0.7	
III: poorly differentiated (%)	31.8	23.7		29.0	26.0		0.6	2.1		0.8	0.9		27.3	30.1		26.9	28.1	
IV: undifferentiated (%)	10.0	8.7		10.2	8.9		31.9	24.9		30	29.9		12.1	8.5		11.6	10.0	
Unknown (%)	53.6	66.0		59.3	63.3		11.4	6.1		7.6	6.6		58.0	56.9		59.4	59.9	
Surgery			—			—			<0.001			0.532			<0.001			0.692
No (%)	0	100		0	100		32.1	12.1		17.5	16.4		20.9	26.6		23.1	24.4	
Yes (%)	100	0		100	100		66.5	86.9		81.8	82.2		76.7	72.5		74.9	74.8	
Radiotherapy			<0.001			0.149			<0.001			0.556			—			—
No (%)	72.3	77.9		80.8	77.2		63.8	91.3		90.4	89.7		0	100		0	100	
Yes (%)	26.7	21.3		19.2	21.7		34.9	8.4		9.2	10.3		100	0		100	100	
Chemotherapy			<0.001			0.864			—			—			<0.001			0.258
No (%)	41.4	16.9		19.7	19.4		0	100		0	100		11.5	43.6		13.8	12.0	
Yes (%)	58.6	83.1		80.3	80.6		100	0		100	100		88.5	56.4		86.2	88.8	
Race			<0.001			<0.001			0.002			0.600			0.022			0.510
White (%)	78.8	76.5		73.6	80.0		77.0	81.0		82.2	81.6		75.5	79.3		79.3	80.6	
Black (%)	12.7	13.3		13.1	11.4		13.6	11.1		10.8	10.0		16.3	11.6		13.0	12.4	
Others (%)	8.5	10.2		13.2	8.6		9.4	7.9		7.0	8.5		8.1	9.1		7.6		
Age of diagnosis			<0.001			0.819			<0.001			0.602			<0.001			0.466
<1 (%)	30.6	36.7		38.5	38.4		27.8	39.9		42.2	42.7		13.9	38.6		16.4	16.1	
1–3 (%)	43.2	41.2		40.3	41.5		47.8	33.3		37.0	37.9		50.3	40.1		53.6	55.6	
4–6 (%)	13.2	10.2		10.6	9.8		13.7	10.3		9.9	9.6		16.9	10.8		14.3	15.2	
7–14 (%)	7.7	5.6		4.7	5.5		6.5	8.4		5.0	4.9		8.8	6.5		7.8	7.4	
15–18 (%)	1.2	1.7		1.4	0.8		1.2	1.5		1.2	0.6		2.2	1.0		1.8	1.1	
>18 (%)	4.1	4.6		4.4	4.0		3.0	6.7		4.7	4.4		7.8	3.0		6.2	4.6	
Year of Diagnosis			<0.001			0.873			0.001			0.622			<0.001			0.884
1975–1984 (%)	14.8	9.2		11.3	11.7		9.6	14.5		11.2	12.4		19.3	8.5		16.7	16.0	
1985–1994 (%)	17.4	12.6		15.6	15.5		13.8	14.0		14.0	14.0		11.3	14.8		12.2	12.7	
1995–2004 (%)	24.0	26.9		26.4	26.3		26.5	24.7		26.0	25.3		22.2	27.1		22.8	24.2	
2005–2016 (%)	43.7	51.4		46.7	46.5		50.1	46.8		48.8	48.3		47.1	49.5		28.2	47.2	
Primary site			0.047			0.101			<0.001			0.805			<0.001			0.124
Adrenal (%)	43.4	39.9		44.8	41.8		49.2	29.4		33.7	35.1		51.4	39.0		51.2	53.9	
Retroperitoneum (%)	11.8	12.3		11.8	10.8		12.6	11.1		13.2	11.7		10.4	12.8		9.1	10.8	
Others (%)	44.8	47.8		43.4	47.8		38.2	59.5		53.1	53.2		38.2	48.2		39.7	35.3	

SG, surgery, CT, chemotherapy; RT, radiotherapy.

**Table 2 jcm-12-00754-t002:** Characteristics of the NB patients and received treatments in the SEER program (1975–2016).

	Patients	
Characteristics	Number	%
Overall	4568	100
Age at diagnosis (years)		
<1	1464	32.0
1–3	1951	42.7
4–6	570	12.5
7–14	327	7.2
15–18	60	1.3
>18	196	4.3
Gender		
Female	2175	47.6
Male	2393	52.4
Race		
Black	582	12.7
White	3581	78.4
Others	405	8.9
Year of diagnosis		
1975–1984	519	11.3
1985–1994	634	13.9
1995–2004	1181	25.9
2005–2016	2234	48.9
Primary site		
Adrenal gland	1931	42.3
Retroperitoneum	552	12.1
Others	2085	45.6
Grade		
I: well differentiated	126	2.8
II: moderately differentiated	52	1.1
III: poorly differentiated	1344	29.4
IV: undifferentiated	436	9.5
Unknown	2610	57.1
Surgery		
No	1147	25.1
Yes	3363	73.6
Unknown	58	1.3
Chemotherapy		
No	1602	35.1
Yes	2966	64.9
Radiotherapy		
No	3354	73.4
Yes	1169	25.6
Unknown	45	1.0

Abbreviations: NB, neuroblastoma; CI, confidence interval; SEER, Surveillance, Epidemiology and End Results.

**Table 3 jcm-12-00754-t003:** Univariate and multivariate Cox regression analyses for predicting the mortality of the patients with neuroblastoma.

	Univariate			Multivariate		
Variables	HR	95% CI	*p*-Value	HR	95% CI	*p*-Value
Gender	1.178	1.058–1.313	0.003			
Grade						
I: well differentiated		1 (reference)	<0.001			
II: moderately differentiated	2.623	1.250–5.502	0.011	2.376	1.098–5.143	0.028
III: poorly differentiated	2.180	1.274–3.730	0.004	2.451	1.396–4.303	0.002
IV: undifferentiated	4.669	2.713–8.033	<0.001	3.693	2.095–6.508	<0.001
Unknown	3.416	2.014–5.792	<0.001	2.304	1.329–3.996	0.003
Radiotherapy	2.269	2.034–2.532	<0.001	1.353	1.201–1.524	<0.001
Surgery	0.392	0.350–0.438	<0.001	0.434	0.384–0.489	<0.001
Chemotherapy	3.452	2.974–4.007	<0.001	2.302	1.945–2.724	<0.001
Race						
White		1 (reference)				
Black	1.233	1.059–1.435	0.007			
unknown	1.141	0.945–1.375	0.171			
Age						
<1		1 (reference)				
1–3	3.927	3.296–4.678	<0.001	3.712	3.091–4.459	<0.001
4–6	4.278	3.478–5.261	<0.001	4.103	3.303–5.097	<0.001
7–14	4.554	3.613–5.740	<0.001	5.222	4.104–6.645	<0.001
15–18	6.135	4.191–8.980	<0.001	6.080	4.124–8.962	<0.001
>18	8.490	6.699–10.760	<0.001	9.783	7.589–12.612	<0.001
Year of Diagnosis						
1975–1984		1 (reference)				
1985–1994	0.622	0.525–0.737	<0.001	0.530	0.442–0.636	<0.001
1995–2004	0.462	0.395–0.539	<0.001	0.354	0.297–0.422	<0.001
2005–2016	0.329	0.283–0.383	<0.001	0.250	0.210–0.297	<0.001
Primary site						
Adrenal		1 (reference)				
Retroperitoneum	0.843	0.714–0.995	0.043	0.741	0.623–0.882	0.01
Others	0.575	0.511–0.646	<0.001	0.526	0.462–0.599	<0.001

**Table 4 jcm-12-00754-t004:** Univariate and multivariate Cox regression analyses for predicting the mortality of patients with neuroblastoma receiving surgery.

Variables	Univariate			Multivariate		
	HR	95% CI	*p*-Value	HR	95% CI	*p*-Value
Gender	1.190	0.932–1.520	0.162			
Grade						
I: well differentiated		1 (reference)				
II: moderately differentiated	3.222	1.335–7.775	0.009	2.847	1.172–6.912	0.021
III: poorly differentiated	2.820	1.446–5.499	0.002	2.691	1.368–5.295	0.004
IV: undifferentiated	5.794	2.952–11.373	<0.001	3.826	1.935–7.568	<0.001
Unknown	3.100	1.602–6.001	0.001	2.326	1.198–4.516	0.013
Radiotherapy	1.916	1.473–2.491	<0.001	1.362	1.169–1.587	<0.001
Chemotherapy	2.714	1.798–4.098	<0.001	2.309	1.784–4.423	<0.001
Race						
White		1 (reference)				
Black	0.877	0.604–1.275	0.493			
Unknown	1.405	1.007–1.962	0.046			
Age at diagnosis (years)						
<1		1 (reference)				
1–3	4.508	3.111–6.531	<0.001	3.382	2.291–4.992	<0.001
4–6	4.778	3.025–7.545	<0.001	3.933	2.448–6.321	<0.001
7–14	5.782	3.390–9.862	<0.001	5.542	3.211–9.567	<0.001
15–18	4.450	1.743–11.363	<0.001	6.304	2.425–16.385	<0.001
>18	9.066	5.512–14.912	<0.001	10.965	6.503–18.489	<0.001
Year of diagnosis						
1975–1984		1 (reference)				
1985–1994	0.803	0.539–1.198	0.283	0.498	0.323–0.768	0.002
1995–2004	0.723	0.501–1.043	0.083	0.408	0.271–0.613	<0.001
2005–2016	0.516	0.358–0.743	<0.001	0.314	0.207–0.478	<0.001
Primary Site						
Adrenal		1 (reference)				
Retroperitoneum	0.739	0.503–1.086	0.124	0.790	0.632–0.986	0.037
Others	0.445	0.340–0.584	<0.001	0.443	0.370–0.530	<0.001

## Data Availability

The data used in this investigation are available to public at https://seer.cancer.gov, https://seer.cancer.gov/data-software/ (accessed on 21 April 2020).

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
