# Peer review of "Long-Term Survival of Neuroblastoma Patients Receiving Surgery, Chemotherapy, and Radiotherapy: A Propensity Score Matching Study"

_jcm, 2023, doi:10.3390/jcm12030754_

Round 1
Reviewer 1 Report
By analyzing the NB patient profiles from SEER database, this manuscript indicates that surgery could improve the long-term survival of NB. Overall, the research is well planned and clearly illustrated. Minor revisions may possibly be needed to improve the quality of this paper.
1. The majority of the NB patients who received surgery could be those of low- and intermediate risk. For those patients, the overall survival rate is quite promising. Yet for high-risk patients, even though they received surgery after the neoadjuvant chemo-induction, currently, their long-term survival rates are not promising. Therefore, it would be more appropriate to re-group the NB patients based on their risk stratification and re-analyze the overall long-term beneficial treatment strategy in each risk group.
2. Immune therapy based on anti-GD2 monoclonal antibodies (dinutuximab, naxitamab) plays a pivotal role in the management of high-risk NB. It would be helpful to include currently available immune therapy in this research, to compare with the conventional treatment methods, regarding to their benefits on patients' long-term survival.
Author Response
Submission to:
Journal name: Journal of Clinical Medicine
Title of paper: Long-term survival of neuroblastoma patients receiving surgery, chemotherapy, and radiotherapy: a propensity score matching study
Manuscript ID: jcm-2025810
Dear editor:
We thank the editors and reviewers for positive comments, excellent and inspiring suggestions regarding our manuscript entitled “Long-term survival of neuroblastoma patients receiving surgery, chemotherapy, and radiotherapy: a propensity score matching study”. The point-to-point responses are embedded in each of the reviewer’s comments and concerns. We made changes accordingly and hope that we have answered all the questions and concerns from the reviewers.
Reviewer #1:
- The majority of the NB patients who received surgery could be those of low- and intermediate risk. For those patients, the overall survival rate is quite promising. Yet for high-risk patients, even though they received surgery after the neoadjuvant chemo-induction, currently, their long-term survival rates are not promising. Therefore, it would be more appropriate to re-group the NB patients based on their risk stratification and re-analyze the overall long-term beneficial treatment strategy in each risk group.
Responses: We appreciate the reviewer’s positive comments and revision guidance on our manuscript. The risk classification for locoregional tumor is based on the International Neuroblastoma Risk Group (INRG) stage, age, resection, biomarkers (MYCN, ploidy, and segmental chromosome abnormalities), and International Neuroblastoma Pathology Classification (INPC), and for metastatic tumors is classified by stage [M (metastatic) and MS (metastatic, special)], age, INPC, and biomarkers (Irwin et al. 2021). While the SEER data lacks relevant information, such as biomarkers and INRG stage, thus we could not evaluate the risk level of patients. Concluding the suggestion of analyzing the overall long-term beneficial treatment strategy in different risk group is essential for this study, we re-analyzed the overall survival of patients in different histopathological grade according to whether receiving surgery, chemotherapy or radiotherapy after propensity score matching study. The results showed significantly longer survival time of patients undergoing surgery in Grade I+II [hazard (HR)= 10.04, p= 0.007] and Grade III [HR= 2.26, p< 0.001], while chemotherapy was unfavorable for prognosis in Grade III group [HR= 0.21, p< 0.001], meanwhile, in Grade I group, chemotherapy had the trend of poor prognosis without significantly statistical difference [HR= 0.30, p= 0.05]. Radiotherapy meant no obvious effect on patients in these two groups. As for patients with advanced histopathological grade (Grade IV), no matter what treatment chosen, a poor long-time prognosis was noted. And we replaced a new figure B for primary figure B.
- Immune therapy based on anti-GD2 monoclonal antibodies (dinutuximab, naxitamab) plays a pivotal role in the management of high-risk NB. It would be helpful to include currently available immune therapy in this research, to compare with the conventional treatment methods, regarding to their benefits on patients' long-term survival.
Responses: Good comments. Considering SEER data did not record whether patients received immune therapy or not, thus we discussed the difference between immune therapy and conventional treatment methods through published literature. Patients with high-risk neuroblastoma have a poor prognosis and survivors are often left with debilitating long-term sequelae from treatment. Prior to 2009, therapy for high-risk neuroblastoma relied on combining surgery, local radiotherapy, and gradually more aggressive combination chemotherapy regimens. While this approach prolonged survival for some, fewer than 40% of patients survived for more than 5 years without relapse; relapsed disease could only rarely be cured (Anderson et al. 2022). After 2010, anti-disialoganglioside-2 (anti-GD2) monoclonal antibody therapy emerged into standard protocol. Although 2-year overall survival of patients receiving adjuvant anti-GD2 monoclonal antibody with interleukin-2 (IL-2), granulocyte-macrophage colony-stimulating factor (GM-CSF) and retinoic acid rose to 86%, while approximately 50% of patients would relapse and eventually died from their disease (Yu et al. 2010), and actual 5-year overall survival rates were only about 50% (Wienke et al. 2021). To pursue more available method for high-risk neuroblastoma, adoptive transfer of chimeric antigen receptor (CAR)-T cells has the potential to build on this success. In early phase clinical trials, CAR-T cell therapy for neuroblastoma has proven safe and feasible, but significant barriers to efficacy remain (Richards et al. 2018).
Reviewer 2 Report
This interesting study examines the impact of radiotherapy, chemotherapy, and surgery on the long-term survival of neuroblastoma patients in the US.
The main result of this study is to display that exclusively surgery has a positive effect on survival from neuroblastoma apart from chemotherapy and radiotherapy which have a negative impact regardless of the stage of the disease.
However, some changes are preliminary to improve the study.
Major comments:
I believe the authors should include the analysis of the relation between the extent of the disease (tumor staging) and survival with the mentioned therapeutic methods in the study.
In Figure 3, grade or stage is shown („different histopathological grades or stages“), but its purpose is unclear (whether it is only the pathohistological or oncological stage of the tumor.
Also, in table 1 it is noted that the authors do not know what the histological type or grade of the tumor is in the 57% of patients.
This means that tumor grade and life expectancy are not related in the vast majority of patients. Additionally, it should be clarified or stated as the primary limitation of the study.
Minor comments:
Page 1 Abstract: should be without headings according to guidelines
Page 1 Keywords: the word „ Surveillance“ should be followed by a comma, not a semicolon
Author Response
Submission to:
Journal name: Journal of Clinical Medicine
Title of paper: Long-term survival of neuroblastoma patients receiving surgery, chemotherapy, and radiotherapy: a propensity score matching study
Manuscript ID: jcm-2025810
Dear editor:
We thank the editors and reviewers for positive comments, excellent and inspiring suggestions regarding our manuscript entitled “Long-term survival of neuroblastoma patients receiving surgery, chemotherapy, and radiotherapy: a propensity score matching study”. The point-to-point responses are embedded in each of the reviewer’s comments and concerns. We made changes accordingly and hope that we have answered all the questions and concerns from the reviewers.
Reviewer #2:
- I believe the authors should include the analysis of the relation between the extent of the disease (tumor staging) and survival with the mentioned therapeutic methods in the study.
Responses: Thank for your comments. We divided the cohorts to three groups according to the histopathological grades (Grade I+II, Grade III and Grade IV), and the calculated survival time undergoing three different treatment. The results showed that significantly longer survival time of patients undergoing surgery in Grade I+II [hazard (HR)= 10.04, p= 0.007] and Grade III [HR= 2.26, p< 0.001], while chemotherapy was unfavorable for prognosis [HR= 0.30, p= 0.05 for Grade I+II, HR= 0.21, p< 0.001 for Grade III] (Wienke et al. 2021) and radiotherapy meant no obvious effect on patients in these two groups. As for patients with advanced histopathological grade (Grade IV), no matter what treatment chosen, a poor long-time prognosis was noted. And we replaced a new figure B for primary figure B.
- In Figure 3, grade or stage is shown (different histopathological grades or stages), but its purpose is unclear (whether it is only the pathohistological or oncological stage of the tumor.
Responses: We are sorry to describe unclearly in Figure 3. The criterion of dividing groups among patients in this study relied on the histopathological grades, which is the only recorded standard of classification, that can evaluate the tumor staging in SEER data. The “stage” in Figure 3 actually represented the histopathological grades of patients, and was corrected to “grade”.
- Also, in table 1 it is noted that the authors do not know what the histological type or grade of the tumor is in the 57% of patients. This means that tumor grade and life expectancy are not related in the vast majority of patients. Additionally, it should be clarified or stated as the primary limitation of the study.
Responses: Thank for your comments. As your concern, the SEER data is incomplete, and misses the grade of approximately 57% of patients. We could only use residual data to group patients, and analyze the different effect of conventional treatment in each group. This deficiency may limit the statistical analysis results and reporting conclusions. We added this deficiency to limitation in article.
- Page 1 Abstract: should be without headings according to guidelines; Page 1 Keywords: the word „ Surveillance“ should be followed by a comma, not a semicolon.
Responses: We corrected these format problems in article.
Round 2
Reviewer 2 Report
In response to my comments, the authors made the necessary adjustments to the paper. I believe the work has been improved enough to merit publishing in the journal.